# Synaptotagmin 1 Suppresses Colorectal Cancer Metastasis by Inhibiting ERK/MAPK Signaling-Mediated Tumor Cell Pseudopodial Formation and Migration

**DOI:** 10.3390/cancers15215282

**Published:** 2023-11-03

**Authors:** Jianyun Shi, Wenjing Li, Zhenhua Jia, Ying Peng, Jiayi Hou, Ning Li, Ruijuan Meng, Wei Fu, Yanlin Feng, Lifei Wu, Lan Zhou, Deping Wang, Jing Shen, Jiasong Chang, Yanqiang Wang, Jimin Cao

**Affiliations:** 1Key Laboratory of Cellular Physiology at Shanxi Medical University, Ministry of Education, and the Department of Physiology, Shanxi Medical University, Taiyuan 030606, China; 2Department of Clinical Laboratory, Shanxi Provincial Academy of Traditional Chinese Medicine, Taiyuan 030071, China; 3Department of Gastrointestinal and Pancreatic Surgery & Hernia and Abdominal Surgery, Shanxi Provincial People’s Hospital, Taiyuan 030045, China; 4Department of Radiology, The First Hospital of Shanxi Medical University, Shanxi Medical University, Taiyuan 030606, China; 5Translational Medicine Research Center, Shanxi Medical University, Taiyuan 030606, China

**Keywords:** synaptotagmin 1, pseudopodial formation, cell migration, metastasis, ERK/MAPK signaling

## Abstract

**Simple Summary:**

Colorectal cancer (CRC) is one major cause of cancer mortality worldwide. Emerging evidence shows that synaptotagmin 1 (SYT1) takes roles in a variety of cancers. However, the role of SYT1 in colorectal cancer remains an enigma. Here, we first assess SYT1 expression levels and discover that its expression is downregulated in CRC tissues and CRC cell lines. We further confirm that SYT1 overexpression suppresses CRC metastasis both in vivo and in vitro using mouse CRC xenograft metastasis model and colon cancer cells. The inhibitory effect of SYT1 overexpression on CRC metastasis is associated with reductions of CRC cell pseudopodial formation, migration, and invasion. Mechanistically, SYT1 overexpression inhibits EMT via negatively regulating the ERK/MAPK signaling, thereby resulting in suppression of CRC cell migration and invasion. Our findings provide new insights into CRC development and indicate the potential of SYT1 as a bio marker and potential therapeutic target for CRC.

**Abstract:**

Although synaptotagmin 1 (SYT1) has been identified participating in a variety of cancers, its role in colorectal cancer (CRC) remains an enigma. This study aimed to demonstrate the effect of SYT1 on CRC metastasis and the underlying mechanism. We first found that SYT1 expressions in CRC tissues were lower than in normal colorectal tissues from the CRC database and collected CRC patients. In addition to this, SYT1 expression was also lower in CRC cell lines than in the normal colorectal cell line. SYT1 expression was downregulated by TGF-β (an EMT mediator) in CRC cell lines. In vitro, SYT1 overexpression repressed pseudopodial formation and reduced cell migration and invasion of CRC cells. SYT1 overexpression also suppressed CRC metastasis in tumor-bearing nude mice in vivo. Moreover, SYT1 overexpression promoted the dephosphorylation of ERK1/2 and downregulated the expressions of Slug and Vimentin, two proteins tightly associated with EMT in tumor metastasis. In conclusion, SYT1 expression is downregulated in CRC. Overexpression of SYT1 suppresses CRC cell migration, invasion, and metastasis by inhibiting ERK/MAPK signaling-mediated CRC cell pseudopodial formation. The study suggests that SYT1 is a suppressor of CRC and may have the potential to be a therapeutic target for CRC.

## 1. Introduction 

Colorectal cancer (CRC) represents a significant health problem as the world’s third most commonly diagnosed and second leading cause of malignancy-associated mortality worldwide [1]. Approximately 9.4% of cancer-related deaths were due to CRC in 2020 and the prevalence of CRC is increasing in recent years [2]. Statistically, 20% of patients diagnosed with CRC have metastasis [3]. Recent studies reported that many factors may contribute to the incidence of CRC, including genetics, diet habits, colon polyp, environment, etc. [3,4]. The carcinogenesis of CRC is a multi-step process which involves a quantity of genomic alterations [5]. If the tumor suppressor gene mutates, the transition from non-invasive to invasive disease may take place. Therefore, it is urgent to explore the roles of these suppressor genes/oncogenes in colorectal cancer for cancer prevention, early diagnosis, and therapeutic development.

Synaptotagmins (SYTs) are a family of structurally related proteins which are highly conserved from invertebrates to human, and expressed in almost all tissues [6]. SYT1 was initially found to work as a Ca^2+^ senser in neurotransmitter release [7]. SYT1 has also been found expressed in non-neuronal cells and plays multiple functions in these cells. For example, SYT1 is required for spindle stability and metaphase-to-anaphase transition in mouse oocytes [6]; SYT1 is expressed in both intestinal epithelial cells and Caco-2BBe cell lines, and is required in cAMP-mediated endocytosis of intestinal epithelial NHE3 cells [8,9], suggesting that SYT1 plays a pivotal role in the physiological activity of the intestinal system. In addition, SYT1 plays important roles in many malignancies. For example, Nord et al. [10] reported that SYT1 is a new oncogene in glioblastoma. Liu et al. [11] demonstrated that dysregulated SYT1 is associated with the survival of head and neck squamous cell carcinoma. Yang et al. [12] found that SYT1 is significantly downregulated in adamantinomatous craniopharyngioma. These studies suggest that SYT1 takes roles in a variety of cancers. However, the potential role of SYT1 in CRC remains unclear.

Here, we comprehensively investigated the expression of SYT1 and its functional roles in the progression of CRC both in vitro and in vivo. Interestingly, we found that SYT1 was downregulated both in CRC tissues and CRC cell lines, and overexpression of SYT1 could suppress CRC cell metastasis in tumor-bearing mice in vivo and migration in vitro. We also demonstrated that SYT1 exerted the above effects through suppressing ERK/MAPK signaling-mediated tumor cell pseudopodia formation. The study may help in understanding the role of SYT1 in the development of CRC.

## 2. Materials and Methods

### 2.1. Public Dataset Acquisition

The gene expression and clinical information data in each tumor and normal sample were obtained from the TCGA database (https://portal.gdc.cancer.gov/, accessed on 17 September 2023). The publicly available TCGA database included 538 tumor tissues, excluding cases with insufficient or missing data on local invasion, lymph node metastasis, distant metastasis, age, overall survival, and TNM staging.

### 2.2. Collection of Human CRC Tissues

CRC tissues and paired adjacent normal tissues were obtained from 15 patients during tumor resection surgeries at the Shanxi Provincial Academy of Traditional Chinese Medicine (Taiyuan, China), and were used to check the expression levels of SYT1. The study was approved by the Medical Research Ethics Committee of the Shanxi Provincial Academy of Traditional Chinese Medicine, Taiyuan, China (approval no.: 2019-06KY005).

### 2.3. Mouse Model of Xenograft CRC Metastasis

The CRC metastatic animal model was constructed by tail vein injection of 1.5 × 10^6^ HCT116 cells in 5-week-old female BALB/C nude mice (Gempharmatech Co., Ltd., Nanjing, China). Mice were kept in a sterile environment during the whole experiment. Eight weeks after tumor injection, the potential tumor metastasis to lung and liver of each mouse was scanned by a microPET-CT (Pingseng Healthcare Co., Ltd., Kunshan, China) after injection of 200 μCi ^18^F-FDG. Maximum standardized uptake (SUVmax) of regions of interest (ROI) were analyzed after manual definition. Mice were then sacrificed and lung and liver tissues were harvested and fixed with 4% paraformaldehyde, embedded in paraffin, and stained with hematoxylin and eosin (H&E). The expressions of Slug and Vimentin were also stained using immunohistochemistry. The animal use protocol was approved by the Animal Ethics Committee of the Shanxi Medical University (Taiyuan, China) (approval no.: SYDL2023-085) and abided by the standards of the National Institute of Health Guide for the Care and Use of Laboratory Animals.

### 2.4. Immunohistochemical and Immunofluorescent Staining

Five-μm tissue sections attached to slides were deparaffinized, blocked, and incubated with primary antibody overnight at 4 °C. After washing with PBS, tissue sections were incubated with HRP-conjugated secondary antibody for 1 h, and, then, the peroxidase activity was visualized by reacting with 3,3-diaminobenzidine (DAB). Tissue sections were washed in water, counterstained with hematoxylin, and cover-slipped. Positive signals were observed and photographed under a microscope. To perform immunofluorescent staining, tissue sections were stained with Alexa Fluor 488 and 594 goat anti-mouse or anti-rabbit IgG (H + L) for 1 h at room temperature. Nuclei were stained with DAPI. The sample was incubated with the primary antibodies SYT1 (1:100), Ki67 (1:400), PCNA (1:400), Slug (1:200), Vimentin (1:200), and p-ERK (1:200). Antibody information is shown in Appendix A.

### 2.5. Cell Lines and Cell Culture

Human colonic carcinoma cell lines SW620, SW480, and HCT116, and the normal colorectal cell line NCM460, were purchased from the Shanghai Institutes for Biological Sciences, at the Chinese Academy of Sciences. SW620 and SW480 cells were maintained in L15 medium (Hyclone Logan, Utah) supplemented with 10% FBS (Gibco Grand Island, NY, USA). HCT116 and NCM460 cells were cultured in DMEM medium (BOSTER) containing 10% FBS. All cells were cultured at 37 °C with 5% CO_2_.

### 2.6. Western Blotting

Western blotting was carried out as we previously described [13]. In brief, cultured cells were collected and washed with PBS, and then were lysed with RIPA protein lysate buffer. The total protein concentration was determined by BCA protein assay kit (Beyotime, Haimen, China). Equal amounts of proteins were separated by SDS-PAGE and transferred onto a PVDF membrane (Bio-Rad, Hercules, CA, USA). Proteins were probed with specific primary antibodies followed by detection with secondary antibodies conjugated with HRP. Signals were detected on a gel imaging system by using ECL (Thermo Scientific, Waltham, MA, USA). The primary antibodies include Slug (1:500), Vimentin (1:500), and GAPDH antibody (1:1000).

### 2.7. RT-qPCR

Total RNA was extracted from SW480 cells and HCT116 cells by TRIZOL reagent (Invitrogen, Carlsbad, CA, USA). RNA extraction and cDNA synthesis were performed according to the manufacturer’s instructions. Then, cDNA amplification was performed by qPCR using Premix Ex Taq™ kit (Takara, Osaka, Japan). qPCR procedures were run at 95 °C for 30 s, followed by 40 cycles at 95 °C for 5 s and, then, 60 °C for 30 s. The primers used in qPCR were designed with Primer 5.0 software and were synthesized with the help of Sangon Biotech (Shanghai, China). Primer sequences are shown in Table 1.

### 2.8. Plasmids and Transfection

The complementary DNA (cDNA) of SYT1 was amplified by PCR and cloned into pcDNA3.1(+) plasmid. The pcDNA3.1(+)-SYT1 (SYT1) and pcDNA3.1(+) (control) plasmids were respectively amplified and purified, and, then, were transfected into SW480 and HCT116 cells using LipoFit 3.0 (Hanbio, Shanghai, China) as per the manufacturer’s protocol. After 48 h transfection, the cells were gathered and employed for further analyses.

### 2.9. Transwell Migration Assay

The transwell system with 24-well polycarbonate membranes and 8 μm pores (Corning Costar, Corning, NY, USA) was used to perform migration assays. The volume of the upper chamber was 200 μL serum-free medium containing 1 × 10^5^ cells. And 600 μL medium supplemented with 20% FBS was added to the lower chamber. The cells were cultured at 37 °C with 5% CO_2_ for 48 h. Then, the wells were removed and gently washed twice with PBS. Subsequently, the wells were fixed with 4% paraformaldehyde for 30 min, and stained with 0.1% crystal violet for 10 min, and, then, images were captured under a microscope.

### 2.10. Wound Healing Assay

Wound healing assay was performed to observe cell migration ability. SW480 and HCT116 cells, which transiently overexpressed SYT1, were seeded in 6-well plates at 5 × 10^5^/well. A single scratch was made across the center of the cell monolayer using a micropipette tip. Then, the cells were washed with PBS to remove cell debris. Images were captured under a microscope at 24 h, 48 h, and 72 h post wounding.

### 2.11. Statistical Analysis

GraphPad Prism 5.0 statistical software was used to perform statistical analysis. Data were presented as mean ± standard deviation (SD) of at least three independent experiments. The two-tailed *t*-test was used for comparison of two groups and one-way ANOVA for multi-group comparison, and *p* < 0.05 was considered statistically significant.

## 3. Results

### 3.1. SYT1 Expression Is Downregulated in Human CRC Tissues and Cell Lines

We first searched the SYT1 expression levels in human CRC tissues and normal colorectal tissues from the TCGA database. Results showed that the mRNA level of SYT1 in CRC tissues were significantly lower than that in the normal colorectal tissues (Figure 1A,B). Results of a Human Protein Atlas (HPA) search showed that the protein expression level of SYT1 in CRC tissues was also lower than in the normal colorectal tissues (Figure 1C).

We then validated the above database search results of SYT1 expression in our collected human CRC tissues and adjacent normal colorectal tissues by immunofluorescent staining. Results showed that SYT1 protein expression was lower in the CRC tissues than in the adjacent normal colorectal tissues (Figure 1D and Appendix A). This result was in concordance with the results of database search.

We further validated the above results of SYT1 expression in three human CRC cell lines (HCT116, SW620, and SW480) and a normal colorectal cell line (NCM460) by Western blotting. Results showed that SYT1 expression was significantly lower in CRC cells than in normal colorectal cells (Figure 1E,F). This cell result was also in agreement with the tissue result, indicating that SYT1 expression is really downregulated in CRC tissues and cells.

Univariate analysis using Cox regression revealed that several factors, including tumor status, distant metastasis, lymph node status, gender, age, and SYT1 expression, were significantly associated with the overall survival of CRC patients (Appendix A).

Taken together, above results suggest that downregulation of SYT1 is closely associated with the malignant progression of CRC; therefore, SYT1 may act as a suppressor gene in CRC.

### 3.2. SYT1 Overexpression Represses Pseudopodial Formation of CRC Cells

Actin cytoskeleton reorganization regulates cell morphological changes, namely, pseudopodium formation, and results in the directional migration and invasion of cancer cells [14,15]. Emerging evidence indicates that epithelial to mesenchymal transition (EMT) is a key driver of CRC progression, and transforming growth factor β (TGF-β) is one of the main mediators of EMT [16,17]. We have shown in Figure 1 that CRC tissues and CRC cells showed a downregulated SYT1 expression compared with respective normal colorectal tissues and cells. We further found that SYT1 expression was downregulated in HCT116 cells after TGF-β stimulation (Figure 2A–C), suggesting that SYT1 may have some effect on EMT. To further explore the role of SYT1 in CRC, we chose SW480 cells and HCT116 cells which had lower SYT1 expression to perform further study. SYT1-overexpressing plasmid was designed and synthesized, and plasmid transfection efficacy was checked in CRC cell lines SW480 and HCT116. Transfection results showed that both the mRNA levels (Figure 2D,E) and the protein levels (Figure 2F,G) of SYT1 in SW480 and HCT116 cells were significantly elevated upon SYT1 overexpression, indicating that the transfection is successful and has high efficacy.

We then checked the effect of SYT1 overexpression on the pseudopodium formation of CRC cells using the plasmid and actin tracker green microfilament green fluorescent probe. The results showed that SYT1 overexpression significantly inhibited the formation of pseudopodia in CRC cells (Figure 2H,I). Because pseudopodial formation is an early sign of tumor cell movement, we speculated that SYT1 overexpression may inhibit CRC cell migration, invasion, and metastasis. The following cellular and animal experiments were performed to validate this speculation.

### 3.3. SYT1 Overexpression Suppresses CRC Cell Migration and Invasion In Vitro

CRC cell migration and invasion abilities were, respectively, examined by transwell assay and wound healing assay. Overexpression of SYT1 substantially reduced the cell motility of HCT116 cells and SW480 cells compared to the control as shown in the transwell assay (Figure 3A,B). Overexpression of SYT1 also inhibited the migration of HCT116 cells and SW480 cells as indicated by the wound closure rates after scratching in the wound healing assay (Figure 3C–E). These results indicate that the expression levels of SYT1 might be negatively correlated with the migration and invasion abilities of colon cancer cells.

### 3.4. SYT1 Overexpression Represses Metastasis of CRC Cells in Mice In Vivo

Above cellular experiments showed that SYT1 overexpression suppressed CRC cell migration and invasion, suggesting that SYT1 may inhibit CRC metastasis in vivo. We thus established a CRC xenograft metastasis nude mice model to examine whether SYT1 overexpression could suppress CRC metastasis at the integrative level. Eight weeks after CRC cell injection into the nude mice, the general information of the mice (including survival) was recorded, and microPET-CT images were taken after tail vein injection of ^18^F-FDG. Results showed that the metastasis-free survival rate of the mice was higher in SYT1-overexpressing mice than in the control mice without SYT1 overexpression (Figure 4A). Consistently, high-metabolism lesions (concentration of ^18^F-FDG) indicative of metastatic tumors were found in the lungs and livers of the control mice, whereas no obvious concentration of ^18^F-FDG was found in the SYT1-overexpressing mice (Figure 4B,C). Histological examination showed a large amount of white clump-like dense tissues (tumors) in the lungs of the control mice, while no obvious clumps were observed in the SYT1-overexpressing mice. In addition, the surface of livers was rough and the elasticity was worse in the control mice compared with the SYT1-overexpressing mice (Figure 4D), which suggest tumor metastasis to the liver. H&E stains showed that the metastatic lesion area of livers and lungs was, obviously, smaller in SYT1-overexpressing mice than in control mice (Figure 4E,F). Immunohistochemical stains of the metastatic lesions in livers and lungs showed that Ki67 and PCNA protein levels were notably lower in the SYT1-overexpressing group than in the control mice (Figure 4G). These findings suggest that SYT1 overexpression weakens CRC metastasis in mice in vivo.

### 3.5. SYT1 Overexpression Downregulates EMT-Associated Slug and Vimentin

The spread of tumor cells is one of the typical behavioral characteristics of malignant tumors. These invasive cells undergo a transformation from an epithelial to a mesenchymal phenotype (epithelial–mesenchymal transition, EMT) [18,19], which plays a crucial role in the initial stage of metastasis. EMT make cells losing their epithelial characteristics such as motility limitation and strong cell-cell junction while obtaining mesenchymal characteristics associated with motility enhancement, cell–cell junction weakening and polarized actin cytoskeleton assembly, resulting in formation of protrusive and invasive pseudopodial structures, so that cells can acquire migration ability and undergo actin cytoskeletal reorganization [20]. Shankar et al. [15] reported that actin-dependent pseudopodial protrusion and tumor cell migration are determinants of EMT. We showed in Figure 2 that SYT1 was negatively associated with the formation of pseudopodia in HCT116 cells and SW480 cells. This phenomenon suggests that SYT1 is potentially associated with the EMT process. We, thus, detected the protein levels of Slug and Vimentin (these two EMT-related proteins) and also SYT1 in the lung and liver tissues of mice. Immunohistochemical stains of lung and liver showed that Slug and Vimentin were lower in SYT1-overexpressing mice than in the control mice, while SYT1 expression level showed the opposite (Figure 5A,B). Western blots of Slug and Vimentin (Figure 5C,D) showed the same trend as the immunohistochemical stains. In line with this, Slug and Vimentin was also downregulated in tumors in situ of SYT1 overexpression group mice (Figure 5E–G), suggesting that SYT1 suppressed the colon cancer cells metastasis.

In addition to this, SYT1 overexpression induced EMT-like cellular morphology in HCT116 and SW480 cells (Figure 6A,B). Furthermore, we detected a decrease in Slug and Vimentin protein levels in HCT116 and SW480 cells of SYT1 overexpression, compared with the control group (Figure 6C–F). Taken together, our results indicate SYT1 participates in inhibiting invasion and metastasis via regulating the EMT process of CRC cell.

### 3.6. SYT1 Overexpression Inhibits EMT via Negatively Regulating the ERK/MAPK Signaling

Although the above results demonstrated that SYT1 overexpression inhibited CRC metastasis both in vitro and in vivo, the underlying mechanism by which SYT1 overexpression suppressed CRC metastasis remained unclear. It is known that ERK/MAPK signaling pathway plays a vital role in the occurrence and development of various malignant tumors, and participates in the regulation of EMT process related to tumor migration [21,22,23]. Slug and Vimentin are two downstream components of the ERK/MAPK signaling pathway [24,25,26,27]. U0126 is a highly selective inhibitor of ERK/MAPK signaling [28,29]. In the present study, we first tested the effect of SYT1 overexpression on ERK1/2 phosphorylation. The results showed that SYT1 overexpression had no effect on the total protein expression of ERK1/2, but significantly decreased the p-ERK1/2 level in HCT116 and SW480 cells (Figure 7A,B). Consistently, SYT1 overexpression decreased the p-ERK1/2 protein levels both in the xenograft metastasis model and in the orthotopic transplantation tumor model (Figure 7C–F). Given the above, we speculated that SYT1 might act as a tumor suppressor via regulating the ERK/MAPK signaling. To test this speculation, we treated HCT116 and SW480 cells with SYT1 overexpression plasmid and the control plasmid plus U0126. Results showed that the inhibitory effect of SYT1 overexpression on cell migration and pseudopodium formation was further enhanced upon U0126 treatment, and this was further supported by wound healing and immunostaining assays (Figure 8A–D). We further confirmed that Slug, Vimentin, and p-ERK1/2 were downregulated in CRC cells under SYT1 overexpression plus U0126 compared to SYT1 overexpression alone (Figure 8E–H). Collectively, our results demonstrate that SYT1 suppresses EMT and CRC cell migration and invasion by negative regulating the ERK/MAPK signaling.

## 4. Discussion

CRC has become a familiar malignant tumor with a high incidence worldwide [30]. Many patients are diagnosed with advanced stage, and recurrence or metastasis may occur after surgery [31]. About 20% of CRC patients suffer from metastatic disease at diagnosis [3]. In spite of recent advances in the management of CRC, metastatic disease remains challenging, and patients are rarely cured. Metastasis is the main cause of death in CRC [32]. Although CRC treatments have been improved, the prognosis remains pessimistic. Therefore, it is of great significance to further investigate the molecular mechanisms of CRC occurrence and metastasis.

CRC is a polygenic disease and is caused by genetic and epigenetic changes in oncogenes, tumor suppressor genes, mismatched repair genes, and cell cycle regulation genes in colonic mucosal cells [33]. Gene mutation is a vital factor in the incidence and development of CRC. For example, high expression and mutation of APC may provide valuable prognostic information for the clinical outcomes of CRC [34]. Similarly, vascular endothelial growth factor plays an important role in CRC and may be used as a prognostic indicator [35]. Furthermore, various types of suppressor genes and oncogenes have been identified related with the diagnosis and prognosis of CRC. Synaptotagmins (SYTs) are a family of synaptic vesicle transport proteins that largely serve as Ca^2+^ sensors in vesicular trafficking and exocytosis [36,37]. SYTs are highly conserved from invertebrates to human and are present in almost all tissues [6]. For instance, SYT1-6 and 9-13 have been detected in brain tissue, while SYT5, SYT9 and SYT13 are also found expressed in β-cells [33]. In addition to this, SYT7, SYT8, and SYT15 are expressed in heart, kidney, and pancreas. Nevertheless, the SYTs family has been gradually uncovered to be connected with human diseases including cancers. SYT7 was demonstrated to play an essential role in non-small cell lung carcinoma, head and neck squamous cell carcinoma (HNSCC), gastric cancer, and CRC [38,39,40,41]. Fu et al. [42] reported that SYT8 promoted pancreatic cancer progression via the TNNI2/ERRα/SIRT1 signaling pathway. Zhang et al. [43] indicated that SYT13 promoted the malignant phenotypes of breast cancer cells by activating the FAK/AKT signaling pathway. These studies suggest a tight association of SYTs with cancers.

Although the roles of SYT1 have been demonstrated in a variety of malignancies, such as glioblastoma and HNSCC [11], the potential role of SYT1 in CRC remains unclear. Here, we comprehensively explored the expression of SYT1 and its functional role in the progression of CRC both in vitro and in vivo. We demonstrated that the expression of SYT1 was significantly downregulated in human CRC tissues compared with the adjacent normal colorectal tissues. We further found that SYT1 expression level was negatively correlated with advanced tumor stage, cervical lymph node metastasis, and advanced clinical stage, suggesting that SYT1 may exert a repressing effect on CRC occurrence and development. Our results strongly suggest that downregulation of SYT1 promotes CRC progression.

Pseudopodial protrusion and the local reorganization of the actin cytoskeleton at the leading edge are related to tumor metastasis [44,45]. A key finding of the present study was that SYT1 could suppress CRC metastasis by inhibiting the pseudopodial formation of tumor cells. We applied a variety of assays to assess the biological functions of SYT1 in CRC metastasis, including pseudopodium formation assessment, wound healing assay, and transwell assay in vitro. In addition, we used xenograft metastasis mice model to evaluate the effect of SYT1 on CRC metastasis in vivo. Both in vitro and in vivo experiments demonstrated that SYT1 repressed CRC metastasis most likely by inhibiting tumor cell pseudopodial formation and migration.

EMT is the basis of tumor cell migration and invasion [18,46]. Vimentin is a mesenchymal marker and plays an vital role in promoting cell migration and is significantly upregulated during tumor metastasis [47]. Slug is also an important factor in promoting tumor cell migration by triggering EMT [48]. We observed that SYT1 expression level affected Vimentin and Slug expressions, suggesting that SYT1 may inhibit CRC cell migration and invasion by regulating EMT. As previously reported, ERK/MAPK signaling pathway is essential for promoting cell proliferation and migration during the occurrence and development of various malignant tumors and participates in EMT regulation process related to tumor cell migration [21,22,49,50,51,52]. Slug and Vimentin are regulated by the ERK/MAPK signaling pathway [24,25,26,27]. Here, we further demonstrated that SYT1 promoted the dephosphorylation of ERK1/2 and decreased the expression of Slug and Vimentin, strongly suggesting that SYT1 overexpression inhibits the migration and invasion of CRC cells by regulating the ERK/MAPK signaling pathway. Taken together, the present study suggests that SYT1 may be a tumor suppressor in CRC. Detailed mechanisms warrant future studies.

## 5. Conclusions

SYT1 expression is downregulated in CRC. SYT1 overexpression suppresses CRC metastasis both in vivo and in vitro. The inhibitory effect of SYT1 overexpression on CRC metastasis is associated with reductions of CRC cell pseudopodial formation, migration, and invasion. SYT1 overexpression can induce ERK1/2 dephosphorylation and result in inhibition of EMT, thereby resulting in suppression of CRC cell migration and invasion. Figure 9 provides a schematic outlining the signaling by which SYT1 suppresses CRC metastasis. Our findings provide new insights into CRC development and indicate the potential of SYT1 as a biomarker and potential therapeutic target for CRC.

## Figures and Tables

**Figure 1 cancers-15-05282-f001:**
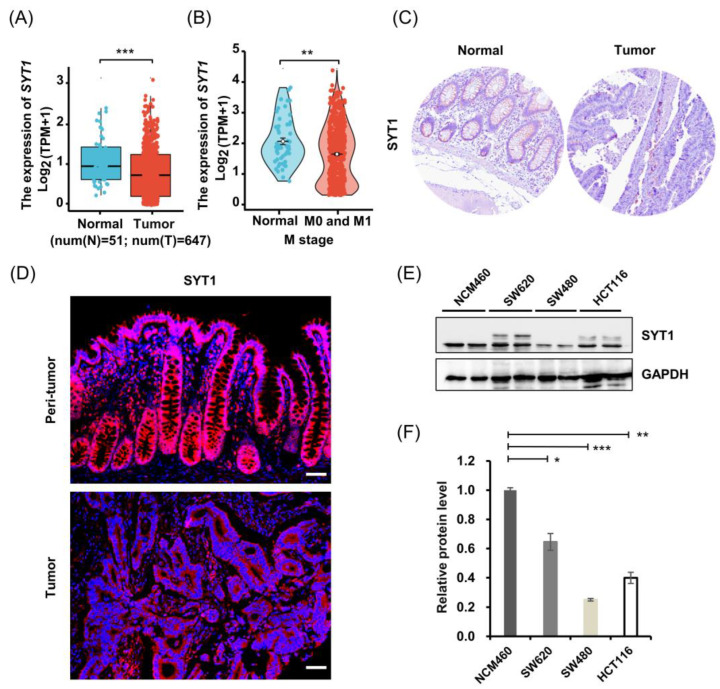
SYT1 was downregulated in human CRC issues and cell lines. (**A**) Relative mRNA levels of SYT1 in CRC tissues and normal colorectal tissues derived from TCGA database; *n* = 51 for normal colorectal tissues and *n* = 647 for CRC tissues. (**B**) Colonic SYT1 mRNA level in CRC patients with distant metastasis (M); *n* = 51 for normal colorectal tissues and *n* = 564 for CRC tissues. N: normal (blue); T: tumor (red). (**C**) Immunohistochemical stains of SYT1 protein in CRC tissues and normal colorectal tissues obtained from HPA database. (**D**) Representative immunofluorescent stains of SYT1 protein in CRC tissues and paired adjacent normal colorectal tissues of the collected CRC cases. Scale bar, 50 μm. (**E**) Western blots of SYT1 proteins in normal colonic cell line NCM460 and CRC cell lines HCT116, SW480, and SW620. (**F**) Grey value statistics of western blots for E; * *p* < 0.05, ** *p* < 0.01, and *** *p* < 0.001. The uncropped bolts are shown in Appendix A.

**Figure 2 cancers-15-05282-f002:**
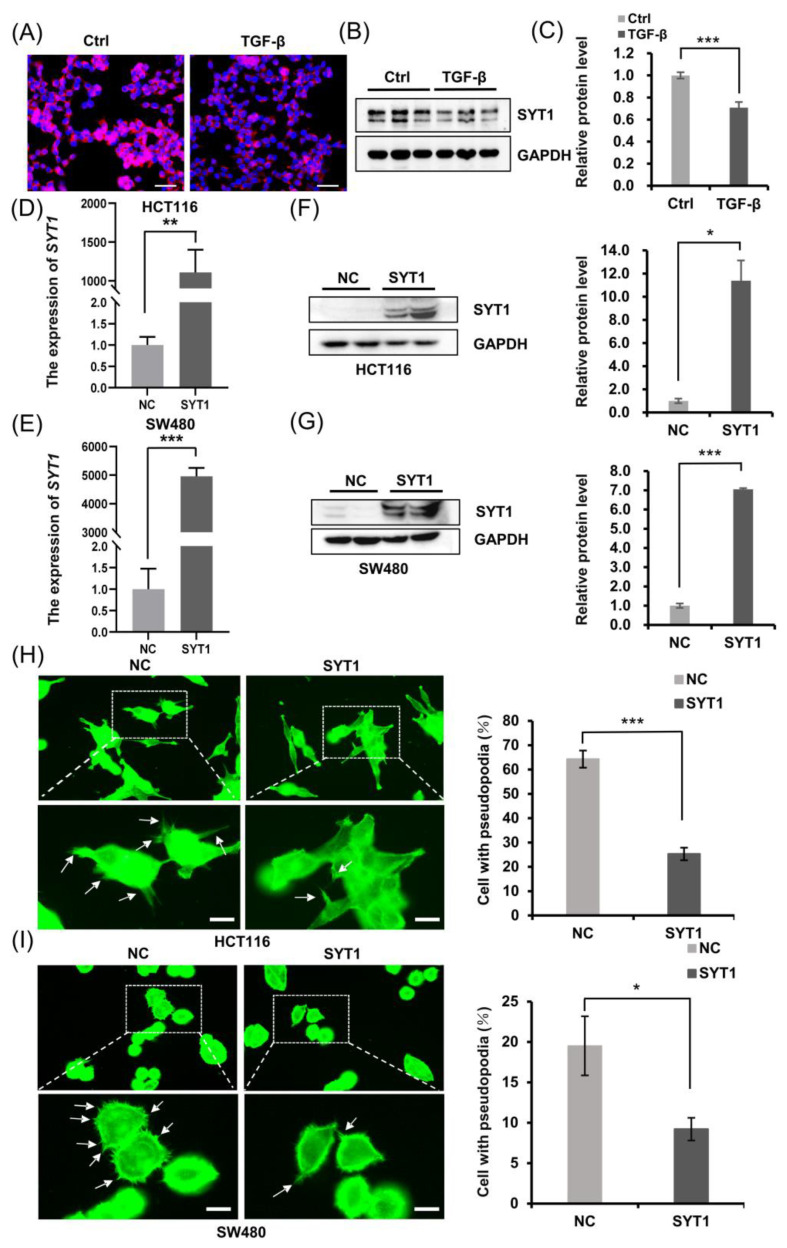
SYT1 overexpression suppressed pseudopodium formation in CRC cells. (**A**) Immunofluorescent stains of SYT1 in HCT116 cells with and without TGF-β treatment. (**B**,**C**) Western blots of SYT1 protein and respective quantitative gray values in HCT116 cells with or without TGF-β treatment. (**D**,**E**) RT-qPCR showing the mRNA levels of SYT1 in HCT116 cells (**D**) and SW480 cells (**E**) with or without pcDNA3.1-SYT1 transfection. (**F**,**G**) Western blots of SYT1 protein in HCT116 cells (**F**) and SW480 cells (**G**) with or without pcDNA3.1-SYT1 transfection. (**H**,**I**) Confocal images of phalloidin (green) showing the pseudopodial formation in HCT116 cells (**H**) and SW480 cells (**I**) with or without SYT1 overexpression. The framed areas were enlarged to better present the pseudopodial protrusions. The white arrow indicates the cell pseudopodium. Scale bar, 25 μm. Quantifications of the pseudopodial protrusions are represented on the right side. Mean ± SD; *n* > 50 cells from three biological repeats; * *p* < 0.05, ** *p* < 0.01, and *** *p* < 0.001. The uncropped bolts are shown in Appendix A.

**Figure 3 cancers-15-05282-f003:**
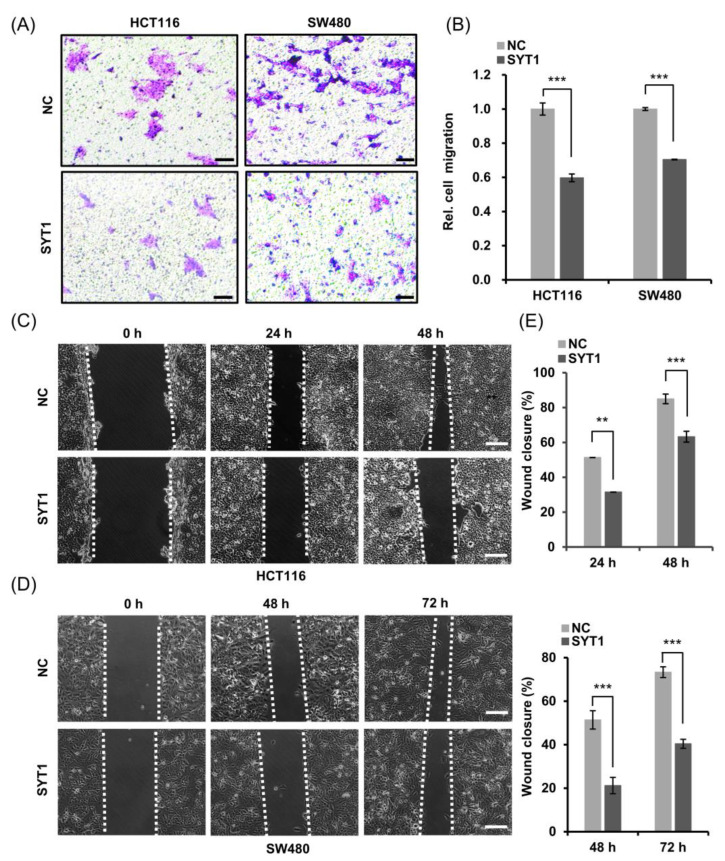
SYT1 overexpression inhibited the migration and invasion abilities of CRC cells in vitro. (**A**) Transwell migration assays of HCT116 cells and SW480 cells with or without SYT1 overexpression. The assays were performed after culture for 48 h. Scale bar, 50 μm. (**B**) Quantitative analysis of migrated cells across the transwell membrane. (**C**,**D**) Wound healing assays showing the invasion abilities of HCT116 cells (**C**) and SW480 cells (**D**) with or without SYT1 overexpression. Scale bar, 50 μm. (**E**) Quantitative analysis of the wound healing rates; ** *p* < 0.01, and *** *p* < 0.001.

**Figure 4 cancers-15-05282-f004:**
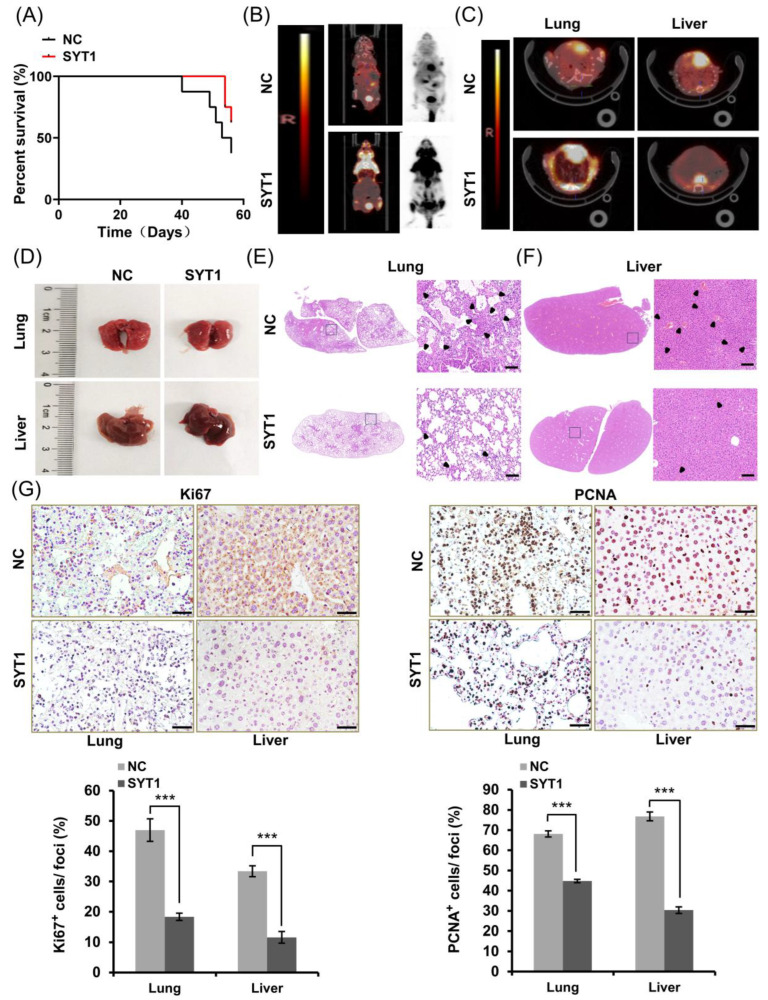
SYT1 overexpression suppressed CRC metastasis in mice in vivo. (**A**) Statistical results of metastasis-free survival rates in control and SYT1-overexpressing nude mice 8 weeks after CRC cell injection. (**B**) PET-CT images of CRC metastasis nude mice model after tail vein injection of ^18^F-FDG. (**C**) PET-CT images of liver and lung of the CRC metastasis nude mice model. (**D**) Gross anatomy of representative lung or liver tissues. (**E**,**F**) H&E stains of lung and liver tissue sections. Framed areas were enlarged to better show the metastases. The black arrowhead indicates the lesion site. Scale bar, 100 μm. (**G**) Immunohistochemical stains of Ki67 and PCNA proteins in the metastatic lesion and precancerous tissue in lungs and livers. Scale bar, 50 μm. *n* = 8 biological replicates for each group; *** *p* < 0.001.

**Figure 5 cancers-15-05282-f005:**
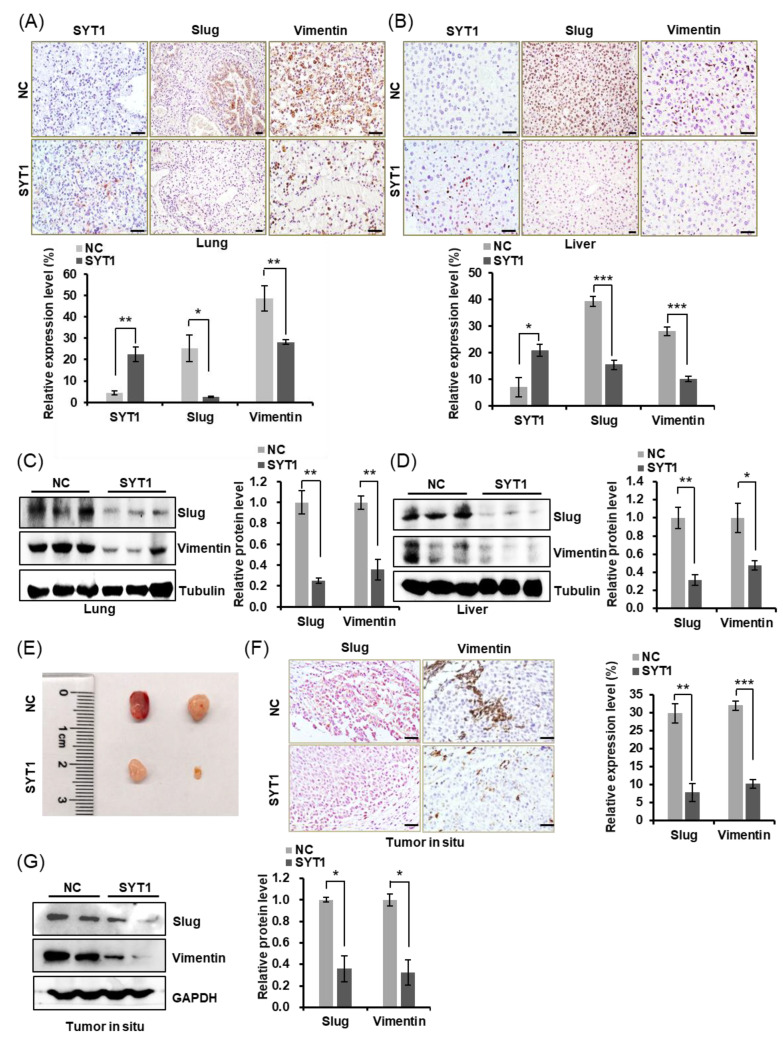
SYT1 overexpression downregulated the expressions of EMT-associated Slug and Vimentin in CRC xenograft metastasis nude mice model in vivo. (**A**,**B**) Immunostaining for SYT1, Slug, and Vimentin in lung (**A**) and liver (**B**) tissues of the control and SYT1 overexpression groups mice metastasis model. Scale bar, 50 μm. (**C**,**D**) The protein levels of Slug and Vimentin in liver and lung tissues of the control and SYT1 overexpression groups mice metastasis model detected by Western blot assays. (**E**) Gross images of tumors in situ of the control and SYT1 overexpression groups mice. (**F**) Immunostaining for Slug and Vimentin in tumors in situ of the control and SYT1 overexpression groups mice. Scale bar, 50 μm. (**G**) The protein levels of Slug and Vimentin in tumors in situ of SYT1 overexpression and the control groups detected by western blot assays. Tubulin or GAPDH was used as the loading control; * *p* < 0.05, ** *p* < 0.01, and *** *p* < 0.001. The uncropped bolts are shown in Appendix A.

**Figure 6 cancers-15-05282-f006:**
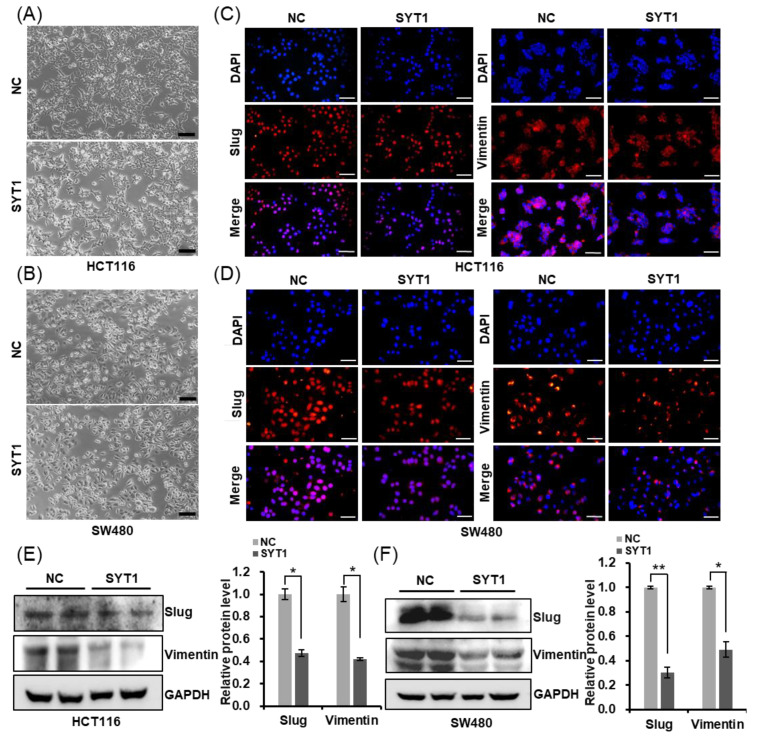
SYT1 suppresses the epithelial–mesenchymal transition (EMT) process of CRC cells in vitro. (**A**,**B**) Morphological changes of HCT116 (**A**) and SW480 (**B**) cells in the control and SYT1 overexpression groups. Scale bar, 50 μm. (**C**,**D**) Immunofluorescent stains of Slug and Vimentin in HCT116 (**C**) and SW480 (**D**) cells at control or SYT1 overexpression. Scale bar, 100 μm. (**E**,**F**) Western blots of EMT markers in HCT116 (**E**) and SW480 (**F**) cells at control or SYT1 overexpression. GAPDH was the loading control; * *p* < 0.05, ** *p* < 0.01. The uncropped bolts are shown in Appendix A.

**Figure 7 cancers-15-05282-f007:**
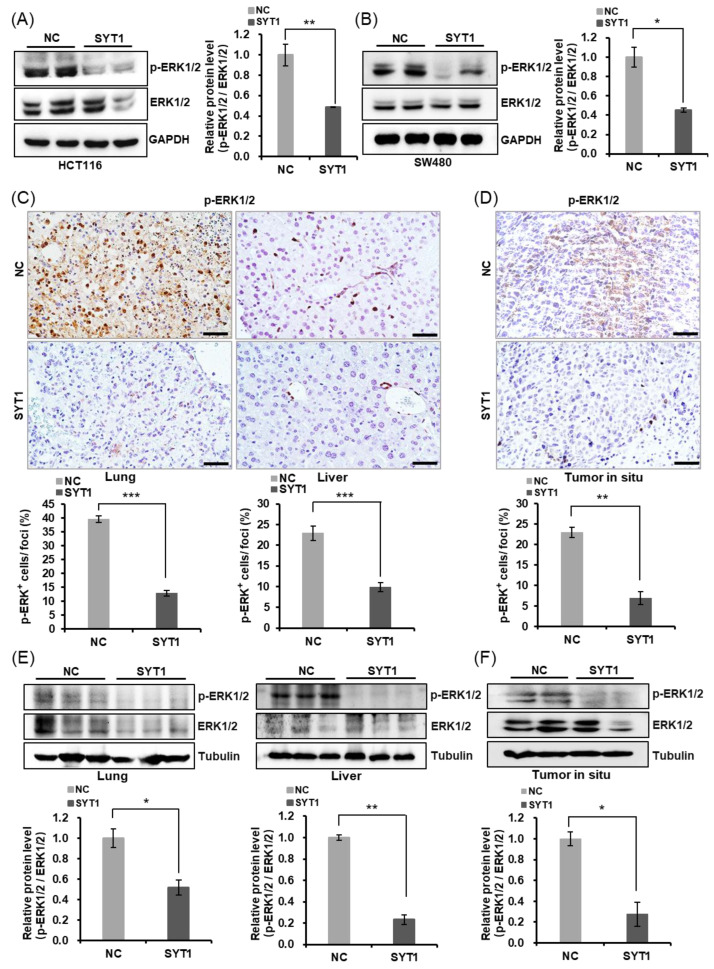
SYT1 inhibited ERK/MAPK signaling. (**A**,**B**) Western blots of MAPK42/44 (ERK1/2) and phosphorylated MAPK42/44 (p-ERK1/2) in HCT116 (**A**) and SW480 (**B**) cells at control or SYT1 overexpression. (**C**) Immunohistochemical stains of p-ERK1/2 in liver and lung tissues from control and SYT1-overexpressing CRC metastasis mice model. Scale bar, 50 μm. (**D**) Immunohistochemical stains of p-ERK1/2 in in situ tumor of control and SYT1-overexpressing mice. Scale bar, 50 μm. (**E**) Western blots of p-ERK1/2 in liver and lung tissues of control and SYT1-overexpressing CRC metastasis mice model. (**F**) Western blots of p-ERK1/2 in orthotopic transplantation tumor of control and SYT1-overexpressing mice. Tubulin or GAPDH was used as the loading control; * *p* < 0.05, ** *p* < 0.01, and *** *p* < 0.001. The uncropped bolts are shown in Appendix A.

**Figure 8 cancers-15-05282-f008:**
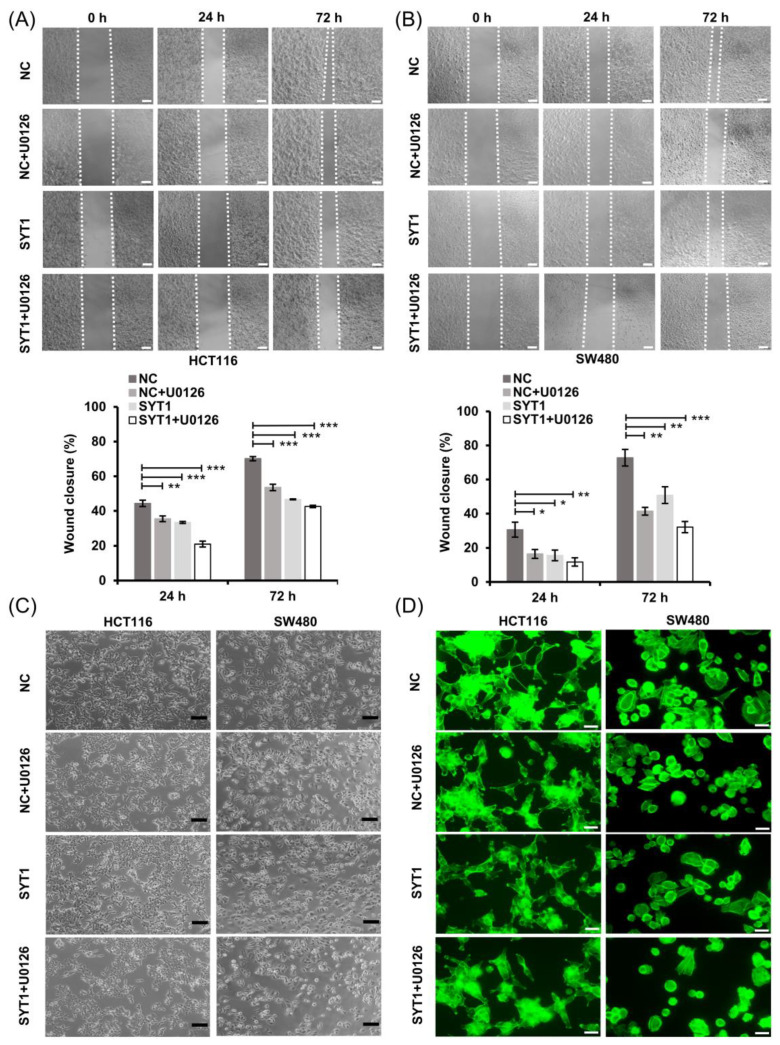
U0126 enhances the inhibitory effect of SYT1 overexpression on EMT and cell migration and invasion. (**A**,**B**) Wound healing assays showing the migration abilities of HCT116 (**A**) and SW480 (**B**) cells in groups of control, SYT1 overexpression, control + U0126, and SYT1 overexpression + U0126. Scale bar, 100 μm. (**C**) Morphological changes of HCT116 and SW480 cells in groups of control, SYT1 overexpression, control + U0126, and SYT1 overexpression + U0126. Scale bar, 100 μm. (**D**) Confocal images of phalloidin (green) in HCT116 and SW480 cells in groups of control, SYT1 overexpression, control + U0126 and SYT1 overexpression + U0126. Scale bar, 50 μm. (**E**,**F**) Immunofluorescent stains of Slug and Vimentin in HCT116 (**E**) and SW480 (**F**) cells of control, SYT1 overexpression, control + U0126 and SYT1 overexpression + U0126 groups. Scale bar, 100 μm. (**G**,**H**) Western blots of SYT1, Slug, Vimentin, and p-ERK1/2 in HCT116 (**G**) and SW480 (**H**) cells of the control, SYT1 overexpression, control + U0126, and SYT1 overexpression + U0126 groups. GAPDH was used as the loading control; * *p* < 0.05, ** *p* < 0.01, and *** *p* < 0.001. The uncropped bolts are shown in Appendix A.

**Figure 9 cancers-15-05282-f009:**
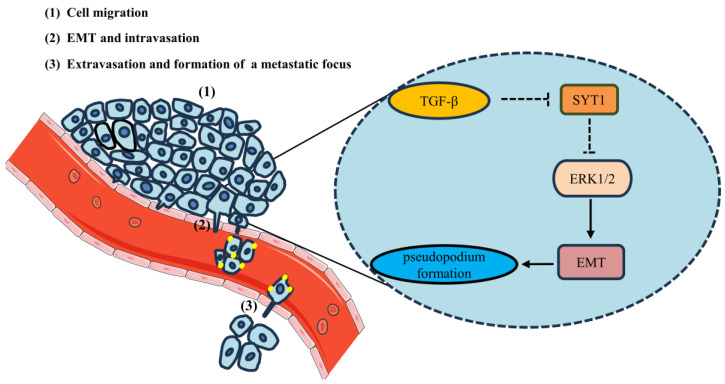
Schematic outlines the signaling by which SYT1 represses CRC cell metastasis.

**Table 1 cancers-15-05282-t001:** Primer sequences.

Gene	Primer Sequence (5′-3′)	
SYT1	Forward	5′-AAAGTCCACCGAAAAACCCTT-3′
	Reverse	5′-CCACCCAATTCCGAGTATGGT-3′
GAPDH	Forward	5′-GGAGCGAGATCCCTCCAAAAT-3′
	Reverse	5′-GGCTGTTGTCATACTTCTCATGG-3′

## Data Availability

The data presented in this study are available on request from the corresponding author.

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
