# Peer review of "Synaptotagmin 1 Suppresses Colorectal Cancer Metastasis by Inhibiting ERK/MAPK Signaling-Mediated Tumor Cell Pseudopodial Formation and Migration"

_cancers, 2023, doi:10.3390/cancers15215282_

Round 1

Reviewer 1 Report

Comments and Suggestions for Authors

This article is about Synaptotagmin 1 and colorectal cancer metastasis. In this comprehensive study, researchers investigated whether the mechanism of Synaptotagmin 1 could inhibit colorectal cancer metastasis using ERK/MAPK signaling-mediated tumor cell pseudopod formation and migration. They screened all colorectal and healthy colon cell lines associated with synaptotagmin 1 expression. They also examined whether overexpressing Synaptotagmin 1 in cells has an effect on EMT and ERK/MAPK signaling in vitro and in vivo. This research article contains very novel findings and is evaluated to reveal Synaptotagmin1 as a new target for colorectal cancer therapy.

Author Response

Responses to Reviewer #1

Comments to the Author:

This article is about Synaptotagmin 1 and colorectal cancer metastasis. In this comprehensive study, researchers investigated whether the mechanism of Synaptotagmin 1 could inhibit colorectal cancer metastasis using ERK/MAPK signaling-mediated tumor cell pseudopod formation and migration. They screened all colorectal and healthy colon cell lines associated with synaptotagmin 1 expression. They also examined whether overexpressing Synaptotagmin 1 in cells has an effect on EMT and ERK/MAPK signaling in vitro and in vivo. This research article contains very novel findings and is evaluated to reveal Synaptotagmin1 as a new target for colorectal cancer therapy.

Response: We thank the reviewer for the interest in our study and we wish that the current version is suitable for publishing in Cancers.

In addition, two corrections were made as follows after carefully checking. The group names of “NC” and “SYT1” were placed in the left side of Figure 5E. And the revised microscopy images for the corresponding tested time point of 48h were provided in the bottom panel of Figure 3C as well. The revised Figure 3C and Figure 5E were provided in the R1 version.

Reviewer 2 Report

Comments and Suggestions for Authors

The authors investigated the role of Synaptotagmin 1 (SYT1) in colorectal cancer metastasis. The aim of the research is clear, as well as the background information on colorectal cancer and SYT1. In general it might be a good idea to give the meaning of an acronym, when first used in the text (e.g. EMT is presented in the simple summary, but its meaning appears later). The methods are described in details, so it is easy to understand. I have a couple of questions about the results:

- Figure !a is confusing to me and I would like to better understand the significant difference between the normal and tumor groups. Based on the figure, it is visible that the SYT1 expression is lower in the tumor group, but I am concerned about the trends. Approximately half of the normal samples showed higher SYT1 expression, but in the other half, tumors showed higher SYT1 expression compared to the surrounding normal tissue. I was wondering if there was any former treatment or other reason, why in half of the cases SYT1 expression decreased compared to the normal tissue, but in the other half it is increased.
- I understand that some of the data came from public datasets, but the authors also collected samples from patients. Did they observe similar phenomenon or in these samples it was clearly decreased SYT1 expression compared to the surrounding normal tissues?
- What is the reason of the presence of multiple bands on the Western blots? It does not seem to be an artifact, because for example on Figure 1E SYT1 has double bands on SW60 and maybe HCT116, but not on the other two. Is it possible that it has more forms and in this case on of the forms are dominant in one sample and the other one is presented in other samples?
- How many images did you use for the statistical analysis of the pseudopodium formation and transwell migration assay? I do see some differences between the NC and SYT1 images, but the difference does not seem to be so significant and it could be that in case you choose another field in the microscopy, there would be even less difference, so the number of images analyzed might be important. (The same with wound healing).
- Figure 4B is confusing to me: maybe I misread the figure, but isn`t this showing that the uptake was higher in the lungs of the SYT1 mice? It seems to be more tumors in the SYT1 mice, than in the NC.
- I do not see very well the SYT1 line in Figure 1A, so I am not sure, if the mice from both groups dies by day 60 or only the negative control mice?

The authors summarized their results and compared with those in literature in the Discussion session, but the answers to the above questions might change the conclusions.

The authors listed 52 references and most of the are from the last 5 years, so the literature presented in the article is up-to-date.

Comments on the Quality of English Language

The language of the paper is understandable, minor correction is needed.

Reviewer 3 Report

Comments and Suggestions for Authors

This article aimed to investigate the role of synaptotagmin 1 (SYT1) in colorectal cancer. The author found that SYT1 is downregulated in CRC tissues and CRC cell lines. SYT1 overexpression suppresses CRC metastasis in vivo and in vitro, which was associated with reductions of CRC cell pseudopodial formation, migration and invasion. Mechanistically, SYT1 overexpression inhibits EMT via ERK/MAPK signaling. SYT1 might be a potential biomarker for future CRC therapies. However, there are some issues needed to be addressed as follows:

1.     It is suggested to add the number of samples to the CRC tissues and normal colorectal tissues derived from TCGA database in Fig. 1A, B.

2.     It is suggested to make statistical analysis and significance analysis on the immunohistochemical results in this paper.

3.     In Fig. 1E, there is one band of SYT1 protein in NCM460 cells, why there are two bands of SYT1 protein in SW620 and HCT116 cells?

4.     It is suggested to normalize the results of mRNA and protein quantitative analysis in this paper.

5.     It is suggested to add the number of samples in Result 4 in the figure legend.

6.     In Fig. 4G, It is suggested to quantify the number of Ki67-positive and PCNA-positive cells.

7.     In Fig. 5E, it is recommended to count tumors in situ and show complete colorectal cancer in situ.

8.     In Fig. 7, a significant difference analysis of p-ERK/ERK ratio should be performed to assess the phosphorylation level of ERK, rather than analyzing p-ERK and ERK separately.

9.     As shown in Fig. 8, it is recommended to detect the overexpression of SYT1 and the inhibitory effect of inhibitor U0126 on ERK signal through Western blot first, and then conduct subsequent wound healing assays and morphological analysis. In addition, it is recommended to explore whether the inhibitor U0126 further suppresses the inhibition of migration and invasion abilities caused by SYT1 overexpression.

10.            Western blot results in Fig. 8G, H did not show bands of SYT1, so the overexpression effect of SYT1 could not be determined.

11.             Does TGF-b induce the mRNA expression of SYT1?

12.            Does SYT1 interacts with ERK1/2?

Round 2

Reviewer 3 Report

Comments and Suggestions for Authors

No more question.